# Gastric Cancer and the Immune System: The Key to Improving Outcomes?

**DOI:** 10.3390/cancers14235940

**Published:** 2022-11-30

**Authors:** Sara H. Keshavjee, Ryan H. Moy, Steven L. Reiner, Sandra W. Ryeom, Sam S. Yoon

**Affiliations:** 1Division of Surgical Oncology, Department of Surgery, Columbia University Irving Medical Center, New York, NY 10032, USA; 2Vagelos College of Physicians and Surgeons, Columbia University, New York, NY 10032, USA; 3Division of Hematology/Oncology, Department of Medicine, Columbia University Irving Medical Center, New York, NY 10032, USA; 4Department of Microbiology and Immunology, Columbia University Irving Medical Center, New York, NY 10032, USA; 5Division of Surgical Sciences, Department of Surgery, Columbia University Irving Medical Center, New York, NY 10032, USA

**Keywords:** gastric cancer, immune checkpoint inhibitors, tumor microenvironment, CAR T cells, CAR-NK cells, antibody-drug conjugates

## Abstract

**Simple Summary:**

Gastric cancer (GC) is a lethal form of cancer usually arising from the inner mucosa layer of the stomach. The treatment options for gastric cancer often includes surgery and chemotherapy, but recurrent or advanced disease remains difficult to cure. In aiming to improve outcomes, newer immune-based therapies using antibodies and strategically altered immune cells are being studied. This review summarizes the immune system’s role in the GC tumor microenvironment as well as the current research on immunologic therapies specific to GC.

**Abstract:**

Gastric adenocarcinoma is by far the most common form of gastric cancer (GC) and is a highly lethal form of cancer arising from the gastric epithelium. GC is an important area of focus of the medical community, given its often late-stage of diagnosis and associated high mortality rate. While surgery and chemotherapy remain the primary treatments, attention has been drawn to the use of immunologic therapies, which have shown promise in the treatment of other malignancies. The role for immune-based therapies has become clearer as we obtain a greater understanding of the role of the immune system in gastric cancer formation and growth. A variety treatment to augment the immune system are under evaluation in clinical trials, and these include immune checkpoint inhibitors, antibody-drug conjugates, and immune cell-based therapies. Here, we review the immune landscape and immune-based therapies for GC.

## 1. Introduction

Gastric cancer (GC) remains a formidable global health problem, with over one million new cases diagnosed worldwide in 2020 (ranking fifth in cancer incidence) and 769,000 deaths worldwide (ranking fourth place in cancer mortality) [1]. Many GC patients present with advanced or metastatic disease due to the insidious nature of common presenting symptoms such as dyspepsia, anorexia, weight loss, and abdominal pain [2]. This delay in presentation contributes to a high mortality rate [3]. Fortunately, the incidence of GC has been declining; however, unexpected populations, such as those under 50 years of age, have seen an increase in rates of GC in studies from the United States, UK, Canada, Chile and Belarus [1,4]. Known risk factors for non-cardia gastric cancer include *Helicobacter pylori* infection, smoking, certain diets rich in smoked and salted foods, pernicious anemia, family history (including known hereditary forms), and prior gastric surgery [5]. The overall declining incidence of GC has often been linked to modifiable risk factors such as decreasing *H. pylori* prevalence, secondary to improved sanitation and use of pharmacologic treatment [4,6]. Research from the Cancer Genome Atlas (TCGA) in 2014 detailed four molecularly distinct categories of GC: Epstein–Barr virus-positive (EBV+), microsatellite instable (MSI), genomically stable (GS), and chromosomal unstable (CIN) [7]. These distinct categories display different anatomical predilections, along with trends towards development in older vs. younger patients [7]. Although some studies hint toward varied response to treatment across subtypes, which will be discussed below, further research is needed to clarify prognostic significance as well as individualized treatment approaches. 

Current standard-of-care treatment for early GC is surgery, while locally advanced GC in many Western countries is often treated with perioperative chemotherapy with fluorouracil, leucovorin, oxaliplatin and docetaxel (FLOT) and surgical resection. The FLOT chemotherapy regimen was established based on the results of the FLOT4-AOI trial which showed superiority over the most commonly used prior regimen of epirubicin, cisplatin, and fluorouracil or capecitabine (ECF/ECX) [8]. In the FLOT group, the median overall survival was 50 months vs. 35 months in the ECF/ECX group. This study only included patients with resectable GC; however, all-comer outcomes are much worse. A recent review of 5-year all-comer survival rates of GC patients between the 1990s and 2010s showed vast differences between countries, with the highest survival rates of 72.1% in Japan, down to 38.4% in America [9]. Some of this difference may be accounted for by stage at presentation, with a much lower percentage of patients having early, localized disease in the American and European studies as compared to countries with GC screening programs such as South Korea and Japan [9]. 

For unresectable or metastatic gastric cancer, median survival in Western countries with medical therapy is only about one year [10]. Chemotherapy often with immunotherapy is the standard first-line approach for the majority of advanced tumors [11]. The addition of immune checkpoint inhibitors (ICI) to chemotherapy may be more beneficial in patients with higher PD-L1 expression. For HER2-positive tumors, the addition of the HER2 antibody trastuzumab to chemotherapy improved survival in the randomized Trastuzumab for Gastric Adenocarcinoma (ToGA) trial [12]. Tumors deficient in one or more mismatch repair proteins generally are microsatellite instability-high (MSI-H) tumors, which appear less responsive to standard chemotherapy and more responsive to ICI [13]. The majority of advanced GC patients’ treatment plans involve chemotherapy and recently it has been suggested that addition of immunologic agents may be clinically indicated to increase overall and progression-free survival, as described below. In aiming to improve outcomes, many recent studies have focused on the use of immune therapies such as ICI and immune cell-based therapies for chemotherapy-refractory gastric cancer, as well as first-line treatment in advanced cases. Since the process of tumorigenesis is closely intertwined with interactions with immune cells [14], the use of various immunologic agents may strengthen the anti-tumor responses of the innate and/or adaptive immune systems. This review focuses on the immune landscape and immune-based therapies for GC, with information obtained from searches in PubMed and ClinicalTrials.gov.

## 2. Immune Cell Landscape: Composition and Prognostic Value

The immune system plays a vital role in the destruction of transformed cells, control of oncogenic pathogens and compounds, and regulation of inflammation [15]. As tumor growth ensues, the release of inflammatory cytokines and mediators triggers vast changes within the tumor microenvironment (TME). Both innate and adaptive immune cells are recruited, including tumor infiltrating lymphocytes (TIL), macrophages, and natural killer cells (NK cells), with varying effects on tumor growth and microenvironment [15]. 

### 2.1. Lymphocytes

Lymphocytes play a key part in the immune landscape of gastric cancer, and they have roles in both anti-tumor immunity and immune tolerance in the TME, depending on the type of lymphocyte and the signaling received. Lymphocytes include all T and B cells originally derived from bone marrow progenitors. Circulating immature precursors of T cells enter the thymus and mature into CD4+CD8+ T cells, which subsequently become selected as single positive CD4+ or CD8+ T cells if they bind self-peptide/MHC complexes with low affinity [16]. Cells that recognize MHC I will become CD8+ T cells with cytotoxic effects on tumor cells and virus-infected cells, while cells that recognize MHC II become CD4+ T cells [17]. CD4+ T cells will further differentiate into various types of helper T cells depending on local signals and T cell receptor interaction strength. These include Th1 (which secrete IFN-γ and TNF-α and activate macrophages), Th2 (which secrete IL-4 and IL-5 and stimulate neutrophil attraction), and Th17 cells (which secrete IL-17) [17,18]. Certain T cells will have a high affinity of self-peptide/MHC complexes, and they may be induced into a regulatory T cell (Treg) phenotype through cytokines [16]. Treg have the ability to cause immune suppression via multiple mechanisms: expression of CTLA-4 to inhibit antigen presenting cells, consumption of IL-2 (which is a pro-inflammatory cytokine), secretion of immune cell inhibitory cytokines (IL-10, IL-35, and TGF-β), and induced immune cell death via granzyme and perforin [16]. Tumor cells themselves can also secrete anti-inflammatory cytokines, such as IL-10 and TGF-β, as well as promote the generation of Treg and produce FAS-ligand to induce apoptosis of activated T cells [19]. In sum, T cells contribute to anti-tumor immunity through CD8+ cytotoxic T cells, but the T cell population may be driven toward tolerance through induction of Treg due to cytokines secreted in the TME.

B cells are another important subset of lymphocytes which have effects on anti-tumor immunity in the TME. B cells mature within lymphoid follicles and can take up antigens via the B cell receptor (BCR) for presentation on MHC II, which can activate helper T cells [18]. This T cell interaction will provide the co-stimulation necessary for proliferation, somatic hypermutation, and class switching of the immunoglobulin constant region of B cell clones [18]. The main function of B cells is antibody production, but they have also been shown to function in antigen presentation and initiation of T cell responses [20].

Multiple subtypes of B cells have been shown to infiltrate the TME, including those for antibody production, those for antigen presentation, and regulatory B cells, which suppress immune responses [21]. B cells have been shown to arrange into tertiary lymphoid structures with noted anti-tumor antibody production in GC samples [22]. Consistent with the role of B cells in anti-tumor immunity, increased CD20+ B cell infiltration in GC has been independently associated with significantly increased overall survival and disease-free survival in a study of 584 GC patients [23]. Specifically, increased numbers of class-switched memory B cells and plasma cells were independently associated with improved prognosis [23]. Single-cell profiling experiments have also shown mucosal associated lymphoid tissue (MALT)-B cells with tertiary lymphoid structures in GC samples [24]. High expression of IgA and complement factors in samples with mature tertiary lymphoid structures was observed, highlighting the possible role of B cell-derived IgA in altering the TME [24]. The samples with mature tertiary lymphoid structures were also associated with increased natural killer T cell (NKT cell) infiltration [24]; these cells have T cell receptors and NK cell receptors with roles in anti-tumor immune response and abundant inflammatory cytokine secretion [25].

There is a close association between B and T cell populations, as B cell antigen presentation can induce CD4+ and CD8+ T cells, and conversely helper T cells may help to mature B cells [23,26,27]. In GC, B cell infiltration has been associated with increased CD4+ and CD8+ T cell infiltration [23]. Other studies have shown that there is a positive correlation between tumor CD8+ T cell infiltration and patient survival in GC; this is theorized to be due to increased anti-tumor immune activity [28,29]. More broadly, a study using TCGA data looked at 22 immune cell types and found that GC samples with high levels of tumor infiltrating lymphocytes (including CD8+ T cells, activated CD4+ memory T cells, follicular helper T cells and pro-inflammatory macrophages) were associated with significantly increased 5-year survival [30]. One meta-analysis found that increased CD3+, CD8+, and CD4+ T cell infiltration was correlated to better overall survival in GC [31]. Although it has been proposed that Treg will cause immune suppression of the TME and worsen GC outcomes, this meta-analysis found that FOXP3+ (Treg) infiltration did not have a clear association with outcomes [31]. Evidently, the role of Treg in prognosis is still incompletely understood, as multiple studies have shown a positive correlation between Treg infiltration and survival [32,33]. Multi-parametric evaluation of the lymphocyte infiltrate across multiple cell sub-types within tumors and lymphoid tissue will likely become an important tool in predicting patient prognosis as more details are learned from ongoing analyses.

### 2.2. Macrophages

Another cell type that is important to the TME is the macrophage. Embryonically derived macrophages, as well as blood monocyte-derived macrophages, form tissue resident macrophages within all organs [34]. Macrophages participate in phagocytosis of foreign and apoptotic cells, as well as debris. They can then present antigens on MHC II molecules for recognition by CD4+ T cells. Macrophages polarize into two unique phenotypes; the inflammatory M1 macrophage (secreting IL-1β, IL-12, TNF-α) and the M2 macrophages characterized by anti-inflammatory signal regulation, such as via IL-10 production [35]. Tumor associated macrophages (TAM) are recruited to the TME, especially in hypoxic conditions, where they then contribute to pro-angiogenic signaling, such as the secretion of VEGF-A [14,36]. The TAM may have qualities that are anti-tumorigenic or pro-tumorigenic. Some TAM maintain a high expression of MHC II with anti-tumor phagocytic activity and inflammatory cytokine secretion to activate anti-tumor responses in adaptive immune cells [34]. Other TAM may be pro-tumorigenic, driven by tumor-secreted mediators to have low MHC II expression and increased CD8+ T cell inhibitor signaling (such as PD-L1, described in detail below) [34].

GC tumor cells have been shown to secrete cytokines that induce macrophage polarization to the M2 phenotype, which aids in creating an immune-suppressive TME [37]. Specifically, the gastric cancer niche is high in IL-6 and IL-8, which may preferentially polarize M2 macrophages [38]. The M2 population may then release TGF-β and IL-10, with roles in inhibiting T cell anti-tumor immunity [37]. These M2 TAM may also play a role in promoting the epithelial to mesenchymal transition of gastric cells [38]. Overall, TAM populations may shift toward M1 vs. M2 predominance, with vast effects on the immune activity of the TME and thus patient prognosis. 

### 2.3. NK Cells

Natural killer (NK) cells are cells that contribute to the innate immune system, with roles in anti-tumor immunity and destruction of microbial pathogens. These cells express killer cell immunoglobulin-like receptors (KIR), which recognize MHC I molecules, leading to inhibition of the NK cell activity [39]. However, NK cells can be activated when they lose inhibitory MHC I dependent signaling, leading them to kill cells that do not express normal levels of MHC I, such as virus-infected cells or tumor cells, which may be missed by MHC I dependent CD8+ T cells [39]. They may also be activated by binding of a ligand to NK activating receptors (NK-AR). 

Studies involving depletion of NK cells or knockdown of key NK cell transcription factors in murine models have shown that NK cells have a role in prevention of various types of carcinomas, as well as prevention of metastasis [40]. In human subjects, studies have shown that the degree of NK cell infiltration is a positive prognostic indicator in many solid tumor types, including GC [40,41,42]. However, the TME may secrete factors such as TGF-β which inhibit NK cell function [40]. In GC, tumor infiltrating NK cells have been shown to have reduced effector function, as they may be inhibited by TAM secretion of factors including TGF-β [41]. One study showed that patients with advanced GC have elevated levels of circulating IL-10 and TGF-β (these may be both tumor- and M2 macrophage-secreted), which is associated with decreased NK cell cytotoxicity [43]. Although NK cells play an important role in anti-tumor immunity, their signaling mechanisms have been shown to be exploited by cytokines in the TME to prevent tumor cell death. 

### 2.4. Clinical Quantification of the Tumor Immune Infiltrate

When analyzing the immune cell infiltrate across patient samples, studies have begun to use the Immunoscore, originally developed for colorectal tumors, which takes into account measurements of density of lymphocytes both at the tumor invasive margin and tumor centers. Cells analyzed include CD3+ T cells, CD45RO+ memory T cells, CD45RA+ naïve T cells, CD57+ NK cells, CD66+ neutrophils, CD68+ macrophages, and others [44]. The Immunoscore has been shown to be of prognostic value, with higher lymphocyte infiltration associated with improved 5-year survival [44,45], especially in those with intestinal histological subtype of gastric cancer [46]. Similarly, the Klintrup–Mäkinen (KM) score, which is a pathological grade from 0–3 of the inflammatory cell infiltration at the tumor invasive margin on H&E stain, has been investigated for prognostic value in GC [46]. A higher KM score is associated with significantly improved 5-year survival across both intestinal and diffuse type gastric cancers [46,47]. When aiming for a less invasive measurement, blood indices of lymphocyte to monocyte ratio and neutrophil to lymphocyte ratio have been shown to correlate to tumor infiltrating lymphocytes, such as exhausted CD8+ T cells, as well as survival in a study of 357 GC patients receiving immunotherapy [48]. 

It is therefore anticipated that the immune cell infiltrate of the TME may be able to help predict patient outcomes. A balance between cell types and immune stimulatory vs. inhibitory signaling, as summarized in Figure 1, may have a vast effect on tumor growth and prognosis. Defining and quantifying the complex GC immune cell infiltrate may help to answer clinical questions such as determining prognosis, predicting chemotherapy response [49], and choosing immunologic agents will be most efficacious in a given patient [15].

## 3. Immune Checkpoint Inhibitors (ICI)

Immune checkpoints are signaling mechanisms that cause immune cell inhibition; the checkpoints can be activated by immune cells themselves or signaling from tumor cells. ICI can prevent immune checkpoints on T cells from signaling, thus promoting greater T cell activation and anti-tumor function [50]. ICI began gaining momentum in oncology immediately following studies in melanoma patients, where substantial improvements in survival were first seen. A prime target for ICI is the PD-1/PD-L1 signaling axis, and both the PD-1 receptor on T cells and the ligand PDL-1 can be targeted. Tumor cells may express PD-L1 as an immune-evasion strategy; PD-L1 on tumor cells or on other immune cells can bind to the PD-1 receptor on T cells, inducing anergy and/or apoptosis [50]. It was originally thought that anti-PD-1 antibodies restore potency to the most dysfunctional and terminally differentiated anti-tumor T cells, which express the highest levels of PD-1. Recent breakthroughs demonstrate that anti-PD-1 antibodies instead induces proliferation and differentiation of self-renewing progenitor TCF1+CD8+ T cells, that paradoxically express intermediate or low levels of PD-1 [51,52,53]. PD-L1 expressing tumors have been linked to poorer outcomes in GC patients [54] but represent a promising population for the use of anti-PD-L1 (atezolizumab, durvalumab, avelumab) or anti-PD1 (nivolumab, pembrolizumab, sintilimab, tislelizumab, retifanlimab) antibody therapies. Another inhibitory checkpoint that can be targeted is CTLA-4, using antibodies such as ipilimumab. CTLA-4 is a receptor found on T cells, which when bound has the ability to dampen T cell activation. CTLA-4 decreases T cell activity both intrinsically, via phosphatase recruitment and inhibition of transcription factors, and extrinsically, by competing with CD28 for costimulatory CD80/86 ligand binding signaling [50]. In addition, CTLA-4 is expressed by Tregs, and therapeutic anti-CTLA-4 antibodies may also promote anti-tumor immunity by depleting Tregs and abrogating their immunosuppressive functions. 

In the Checkmate 649 phase III trial, an anti-PD1 antibody, nivolumab, combined with chemotherapy was shown to improve overall survival in patients with PD-L1 combined positive score (CPS) of both ≥1% and ≥5% vs. chemotherapy alone. This led to FDA approval of nivolumab for use in advanced or metastatic GC or gastro-esophageal junction cancer (GEJC) regardless of PD-L1 CPS [55]. Nivolumab has also been shown to prolong disease-free survival in the adjuvant setting, as shown in the phase III Checkmate 577 study [56]. This trial enrolled patients with esophageal or GEJC who had residual disease after chemoradiation. Another anti-PD1 antibody, pembrolizumab, has been shown in the Phase III KEYNOTE-062 trial to be no less effective than chemotherapy as first line treatment in advanced GC/GEJC with CPS ≥1, with fewer treatment-related adverse events than chemotherapy [57]. This trial also showed that overall, the combination of pembrolizumab plus chemotherapy was not superior to chemotherapy alone; however, post hoc analysis of this trial, along with KEYNOTE-59 and KEYNOTE-61, showed a significant benefit for pembrolizumab in the subset of patients with MSI-high tumors [58]. The subsequent KEYNOTE-158 Phase II trial of pembrolizumab for non-colorectal MSI and mismatch repair deficient (dMMR) tumors, which included a subset of patients with GC, led to the approval of pembrolizumab by the US FDA for the use in dMMR and MSI solid tumors [59]. 

One meta-analysis has helped to analyze the randomized controlled trials that have studied first-line advanced GC treatment with chemotherapy plus ICI vs. chemotherapy alone [60]. This study found a significant reduction in death and progression risk in those treated with combination chemotherapy plus ICI, across patients with PDL-1 CPS ≥ 10 and CPS ≥ 1 [60]. Of note, a larger reduction in risk of death and progression was seen in patients with CPS ≥ 10 [60]. In addition to defining the population of GC patients that will most benefit from ICI, studies must evaluate the timing of adjuvant and/or neoadjuvant ICI therapy. Recruitment is currently underway for the MATTERHORN phase III study (NCT04592913) of neoadjuvant durvalumab (anti-PDL-1) or placebo and FLOT chemotherapy followed by adjuvant durvalumab or placebo monotherapy in patients with resectable GC/GEJC, with results expected in 2025 [61]. 

Other studies have also evaluated the use of anti-CTLA-4 antibodies, often in combination with other ICI such as anti-PD1, in comparison to conventional chemotherapy. The Phase I/II CHECKMATE-032 study evaluated nivolumab versus nivolumab plus ipilimumab and showed anti-tumor activity with acceptable safety profile in chemotherapy-resistant patients [62]. The greatest objective response rate was observed in the group treated with nivolumab 1 mg/kg and ipilimumab 3 mg/kg, but there was not a significant increase in overall survival versus the other groups [62]. This led to the inclusion of ipilimumab in the CHECKMATE 649 trial (NCT02872116) of chemotherapy vs. nivolumab and chemotherapy vs. nivolumab plus ipilimumab as first line treatment in advanced GC patients. Although nivolumab plus chemotherapy showed significantly improved overall and progression-free survival (PFS) compared to chemotherapy alone, results on the nivolumab plus ipilimumab arm of the trial are still forthcoming. Another study of interest is currently recruiting patients to investigate safety, PFS and duration of remission with combined ipilimumab, pembrolizumab and durvalumab in patients with multiple solid tumors, including GC (NCT05187338). Adding to the knowledge on combination therapy, an ongoing phase I trial is studying XmAb20717 (monoclonal antibody targeting both PD-1 and CTLA-4) in patients with advanced solid tumors, including GC (NCT03517488). 

Current studies are also focusing on combination therapy with ICI, for example a dual target anti-TGFβ and anti-PD-L1 antibody (M7824) in combination with paclitaxel for second line treatment of recurrent or metastatic GC (NCT04835896). TGF-β is a promising additional target, as it has been known to play a role in immunosuppression within the TME [63], including the activation of Tregs and suppression of activated T cells and NK cells [64,65]. TGF-β neutralization has been shown to increase NK cell anti-tumor activity in vitro [66]. TGF-β also contributes to tumor stromal expansion, which is associated with worsened overall survival in GC [63,67]. ICI treatment may also be combined with anti-VEGF-A antibodies or small molecule inhibitors. VEGF-A is a local mediator secreted by cells in hypoxic environments, often leading to formation of poor-quality vasculature [68]. In addition to promoting tumor growth via formation of vasculature, VEGF-A has also been shown to promote an immunosuppressive TME via upregulation of Tregs, promotion of M2 TAM [68], and enhancement CTLA-4 and PD-1 signaling for suppression of CD8+ T cells in murine models [69]. Results of a clinical trial that will evaluate the use of durvalumab (anti-PD-L1 antibody) with cabozantinib (a multi-tyrosine kinase inhibitor of VEGFR2, MET, RET, AXL, KIT, and FLT3) in GC patients (NCT03539822) are anticipated. Another trial is underway to evaluate the efficacy of cabozantinib with pembrolizumab in recurrent or metastatic GC patients (NCT04164979). Further clinical trials of interest utilizing ICI in GC patients are listed in Table 1. 

### ICI and HER2 Blockade

A minority of GCs show HER2 overexpression, which drives pathways such as PI3K/Akt/mTOR and MAPK [70]. While HER2 overexpression was initially recognized in subsets of breast cancer, HER2 overexpression is now used for targeted treatment of gastric, colorectal, bladder, and pancreatic cancer, among others [70]. Trastuzumab became first line therapy in the treatment for HER2+ GC after the phase III ToGA trial [12]. This study showed increased overall survival of 13.8 months vs. 11.1 months (*p* = 0.0046) in those with HER2+ GC treated with trastuzumab in combination with chemotherapy vs. chemotherapy alone, leading trastuzumab to become standard-of-care for advanced HER2+ GC patients. Subsequent studies have examined the combination of trastuzumab with ICI. Preliminary results of a phase III study on pembrolizumab and trastuzumab plus chemotherapy vs. trastuzumab and chemotherapy (NCT03615326) demonstrated an increased overall response rate when pembrolizumab was added [71]. The addition of ICI may be of particular importance as we aim to reduce adverse event rates common to GC chemotherapy regimens, which often include a fluoropyrimidine plus platin therapy [72]. The MAHOGANY trial (NCT04082364) is currently evaluating the use of margetuximab (a monoclonal Fc-optimized HER2 antibody) plus retifanlimab (an anti-PD-1 antibody) with and without chemotherapy in unresectable GC and GEJC [73]. Initial reports suggest a favorable safety profile [74], and additional results on efficacy for this novel chemotherapy-free combination regimen are pending.

## 4. Antibody-Drug Conjugates

Antibody drug conjugates (ADC) target a cytotoxic drug to a specific cell type via antibody-antigen interaction. When an ADC binds to a target antigen, it enters via receptor-mediated endocytosis; intracellular proteases can then release the conjugated drug [75]. In addition to individual tumor cell death via cytotoxic drug release, neighboring cell death may occur via drug diffusion, and widespread cell death may ensue secondary to local inflammation from release of damage-associated molecular patterns [75]. ADC featuring an anti-HER2 antibody have been most studied in GC patients, specifically the subset who are HER2+. The conjugation of trastuzumab and the topoisomerase inhibitor deruxtecan, altogether called DS8201-a or Enhertu^®^, is currently being investigated for use in breast, colon and gastric cancer. A clinical trial of this drug vs. chemotherapy alone (NCT03329690) showed significantly increased objective response rate as well as overall survival in HER2+ GC patients who had progressed on at least two prior therapies [76]. Notably, a larger percentage of patients discontinued or decreased dosage of therapy in the Enhertu^®^ group than the chemotherapy group [76]. Along with myelosuppression, interstitial lung disease was a noted issue in 10% of patients [76]; this has been previously associated with other anti-HER2 and topoisomerase inhibitor therapies [77]. After further clinical trials showed favorable responses in HER2+ GC and GEJC patients, Enhertu^®^ was approved by the US FDA in January of 2021 [78]. Another ADC named RC48 consists of hertuzumab and monomethyl auristatin E (MMAE), which inhibits tubulin polymerization. RC48 has shown tolerable safety and antitumor activity in phase I/II trials of HER2+ solid tumor patients, with more clinical trials underway [78].

Another target antigen for ADC is TROP2, the tumor-associated calcium signal transducer 2. TROP2 is a transmembrane calcium signal transducer that drives cell proliferation and formation of metastasis. This protein is overexpressed in multiple tumor types including some GC, with limited expression in healthy tissue, making it a good target for ADC [79]. The IMMU-132-01 phase I/II basket trial (NCT01631552) treated patients with metastatic epithelial tumors (including GC) with sacituzumab govitecan (anti-trop-2 monoclonal antibody coupled to SN-38, the active metabolite of the topoisomerase I inhibitor, irinotecan) [80]. They found a tolerable safety profile compared to similar agents and promising efficacy in certain tumors [80]; however, a very small percentage of participants were GC patients, and we await further trials on this ADC.

Despite promising options for antibody-based therapies, research on which therapy may be best suited for which patient is still ongoing. Biomarkers that may be used to classify tumors, as well as possibly affect treatment decisions and prognosis, include mismatch repair status, MSI identification, PDL-1 combined positive score (CPS), tumor infiltrating lymphocyte quantification, and tumor mutational burden [81]. Interestingly, the use of circulating tumor DNA (ctDNA) has the potential to provide data on likelihood of response to ICI and prognosis without requiring a tissue biopsy [82]. Ongoing research will be necessarily to validate various biomarkers for clinical use in prognosis, treatment selection and evaluation of treatment response.

## 5. CAR-T Cells

Many experts predict the future of cancer treatment to be patient- and tumor-specific. The advent of autologous T cells engineered to express specific chimeric antigen receptors (CARs) has the potential to individualize cancer treatment. These CAR-T cells can be customized to essentially any tumor-associated antigen and can be adapted with co-stimulatory domains to further increase CAR-T cell responses to eliminate tumor cells [83]. In order for CAR-T cells to be generated, leukocytes are taken from the patient’s blood, and enriched for T cells. These T cells are cultured and further enriched for specific populations, as well as activated with growth factors such as IL-2 [84]. CARs are introduced to the T cells, usually via viral vectors, and the cells are expanded in a reactor until the population size is suitable for patient use [84]. Before a patient is given their CAR-T cells, they must be treated with lymphodepleting chemotherapy (usually fludarabine and cyclophosphamide) to deplete native Tregs and other immune cells and thus improve the anti-tumor function of infused CAR-T cells [85].

CAR-T cells were initially used with a target to CD19 for chronic lymphocytic leukemia, but CAR T-cells have been challenging to adapt for use on solid tumors [86]. Solid tumors confer a physical barrier to T cell entry due to enhanced stroma tissue and unconventional tumor vasculature [87], along with a hypoxic environment with significant oxidative stress that does not favor high T cell activity [83]. Solid tumors also take advantage of immunosuppressive cytokines and checkpoint signaling (PD-1/PD-L1 and CTLA-4) to decrease T cell activity, creating increased challenges for CAR-T cells [83]. Additionally, some tumors are able to change their antigen expression based on immune cell population pressure, which may be overcome with CAR-T cells with dual-CARs [88]. Adverse events have also been a barrier to use, including cytokine release syndrome (CRS). CRS following CAR-T cell therapy presents with symptoms such as fever, rash, nausea/vomiting, diarrhea, tachycardia, hypotension, and neurologic symptoms [89]. CRS is caused by the antigen-induced activation of a large quantity of T cells after infusion; the speed of onset of CRS is thought to be correlated to number of T cells infused and the severity to the tumor burden [89,90]. This syndrome is driven by many cytokines, but the main effects are thought to be due to IL-6 and its subsequent pro-inflammatory effect on many cell types [89]. Treatment involves the use of tocilizumab (anti-IL-6 monoclonal antibody) and/or corticosteroids [85].

When selecting a CAR for T cells targeting a specific tumor type, the antigen must be tumor-specific to avoid toxicity. One promising target for CAR T cells in GC is claudin 18.2. This protein is selectively expressed in tight junctions of gastric mucosa, making this target useful for its specificity [91]. Phase I interim results of a study using claudin18.2-specific CAR-T cells (NCT03874897) have shown an acceptable safety profile in claudin18.2 positive GC, with overall response rate of 57.1% and disease control rate of 75% [91]. CRS classified as grade I (not life threatening, requires symptomatic treatment only) or II (requires and responds to moderate intervention; oxygen requirement < 40%, hypotension responsive to fluids or low dose of one vasopressor or grade 2 organ toxicity) was seen in 94.6% of patients, but no grade III or higher instances of CRS were seen and there were no dose-related toxicities [89,91].

Another phase I trial is underway evaluating CEA targeted CAR-T cells in CEA positive malignancies including colorectal, esophageal, gastric and pancreatic malignancies (NCT05415475). Also of interest, a phase I/II trial is evaluating HER2-specific dual-switch CAR-T cells, BPX-603, administered with rimiducid to subjects with previously treated, locally advanced or metastatic solid tumors, including GC, which are HER2 amplified or overexpressed (NCT04650451). Another phase I/II trial will be evaluating the safety and efficacy of mesothelin-specific CAR-T cells in the treatment of GC (NCT03941626) [92]. We await further results of these trials, among others, to determine the role for CAR-T cells in the treatment of GC.

## 6. CAR-NK Cells

NK cells are innate immune cells that have the ability to maintain pathogen-specific memory and also to kill tumor cells [93]. They also produce immune-modulating cytokines, such as IFN-γ and TNF-α [93]. Studies have shown that NK cells in GC patients are both decreased in number and in cytotoxic cytokine secretion ability compared to healthy controls [94]. The percentage of NK cells in peripheral blood is also positively correlated to prognosis in GC patients [95]. Harnessing the useful anti-tumor properties of NK cells, CAR-NK cells have shown promise for cancer treatment due to their ability to target specific tumor cells via CARs with lower risk of complications such as graft-versus-host disease, neurotoxicity, and CRS than are seen with the use of CAR-T cells [96]. CAR-NK cells kill tumor cells through CAR-dependent mechanisms, but they can also kill tumor cells through CAR-independent mechanisms (such as through natural cytotoxicity receptors), which is especially useful in heterogeneous solid tumors [96]. Additionally, they contribute to tumor cell death via antibody-dependent cell-mediated cytotoxicity (via the Fc receptor CD16); this may be exploited by the combination of NK cells and tumor-specific antibodies [96].

NK cells have the advantage of potential for commercial use, as they can be given in allogeneic infusions as “off the shelf” products. These cells are produced by expanding cell lines such as NK92 cells, or isolation from peripheral blood mononuclear cells (PBMCs), umbilical cord blood, CD34^+^ hematopoietic progenitor cells (HPCs), and induced pluripotent stem cells (iPSCs) [96]. These cells can be made into CAR-NK cells by retroviral or lentiviral transduction, and ongoing research is investigating the use of transposon systems and CRISPR/Cas9 to introduce CARs [96]. Techniques for efficient transduction with long-lasting CAR expression as well as for efficient means of NK cell expansion will need to be optimized before commercial application of CAR-NK cells becomes a reality.

Commercial “off the shelf” CAR-NK cells have been given in allogeneic infusions for hematologic cancers and are now being studied for use in patients with various solid tumor types. Before administration of CAR-NK cells, patients are pre-treated with lymphodepleting chemotherapy (usually fludarabine and cyclophosphamide), which has been shown to allow a longer lifespan of infused NK cells [96,97]. In the largest clinical trial utilizing CAR-NK cell treatment with published data (NCT04245722), preliminary data reported no cases of graft-versus-host disease or immune effector cell-associated neurotoxicity syndrome in 20 treated patients. Only two cases of CRS (1 grade I, 1 grade II) were seen [98], which is much less frequent than with CAR T cells as mentioned above. Main adverse events such as neutropenia, anemia and thrombocytopenia may be related to the lymphodepleting chemotherapy given before infusion.

Mouse models of gastric cancer have shown promising anti-tumor activity of CAR-NK cells, such as with mesothelin-specific CAR-NK cells [99]. A study by Cao et al. showed significant anti-tumor activity of NK-92 cells in mouse-implanted, patient-derived GC xenografts [99]. These NK cells exhibit potent anti-tumor activity due to a genetically engineered decrease in inhibitory receptors and are promising for widespread use, as they are derived from a cell line that can be commercially expanded ex vivo [100]. Several clinical trials utilizing NK cells and CAR-NK cells are ongoing in patients with solid tumors, including GC, as summarized in Table 2.

## 7. Conclusions

The role of the immune system in GC is complex, as the balance between immune inhibiting and immune activating signals can alter tumor growth and thus patient prognosis. Although further research on the immune infiltrate will be necessary, existing knowledge has outlined target pathways for immune-based therapies to alter the balance toward anti-tumor immune function. Antibody-based therapies can inhibit signaling pathways such as immune checkpoints and improve survival in select patients. As ICI becomes standard of care, attention must also be given to methods of augmenting tumor cell death via complementary pathways. Cell-based therapies, such as CAR T and CAR NK cells are also a promising area of research, and further information on their utility in GC is anticipated.

This review highlights what is currently known about the gastric cancer immune microenvironment and immunotherapies. Where we go from here is difficult to predict. Surely, we will continue to learn more about the immune microenvironment and how to manipulate it. However, whether we will make faster progress in improving ICI, antibody-based therapies, cell-based therapies, or a new form of immunotherapy is unclear. Precision medicine is the future of oncology, and one can anticipate that care will be driven by immunotherapies customized to an individual’s tumor biology. Periodic genetic analysis of a given tumor, whether through biopsy or ctDNA, will enable oncologists to analyze a tumor’s biological evolution as it relates to treatment efficacy. This will then allow the timely alteration of patient-specific cancer treatment throughout a disease course to improve outcomes and confront treatment resistance. There is much research in this area that remains. However, in the words of Abraham Lincoln: “The best way to predict the future is to create it.”

## Figures and Tables

**Figure 1 cancers-14-05940-f001:**
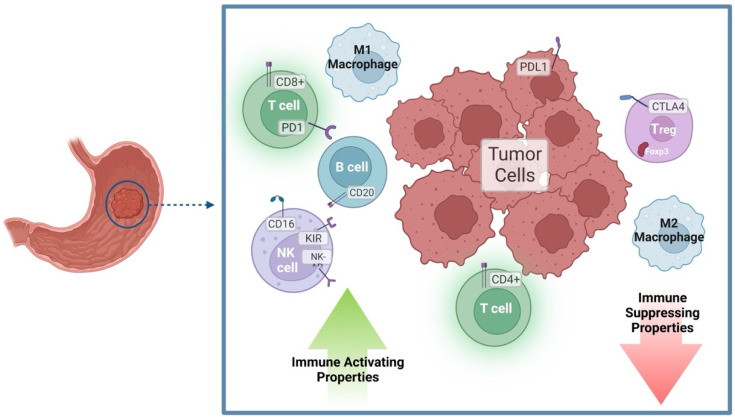
Infiltrating immune cells balance the inflammatory vs. immuno-suppressive tumor microenvironment (TME) in GC. CD8+ T cells are generally cytotoxic to tumor cells, although the tumor may evade the immune system via PD-1/PD-L-1 signaling. CD4+ T cells may differentiate into T helper subtypes with pro-inflammatory properties. However, differentiation into Treg with CTLA4 expression will drive an inhibition of immune cell activity. B cells are generally thought to have anti-tumor immune activating functions. M1 macrophages are pro-inflammatory and drive immune activation in the TME, while M2 macrophages decrease immune responses. NK cells may have anti-tumor properties which can be inhibited by tumor-secreted factors. Figure created with BioRender.com (accessed on 25 October 2022).

**Table 1 cancers-14-05940-t001:** Select Ongoing Trials Involving Immune Checkpoint Inhibitors for GC.

Clinical Trial Information	Immunotherapy Type	Other Therapy	Patient Population	Status
**NCT04592913** **MATTERHORN Phase III**	Neoadjuvant-Adjuvant Durvalumab vs. Adjuvant Durvalumab	Neoadjuvant-Adjuvant FLOT chemotherapy	GC and GEJC	Recruiting
**NCT02872116 Checkmate 649, Phase III**	Nivolumab Plus Ipilimumab vs. Nivolumab in Combination with chemotherapy vs. chemotherapy alone	Oxaliplatin + Leucovorin + Fluorouracil (FOLFOX) or Oxaliplatin + Capecitabine (XELOX)	Previously untreated advanced or metastatic GC or GEJC	Active, not recruiting
**NCT03517488 Phase I**	XmAb20717 (monoclonal antibody targeting both PD1 and CTLA-4)	N/A	Multiple tumor types, including GC	Active, not recruiting
**NCT04835896 Phase Ib/II**	M7824 (Bintrafusp Alfa, dual target anti-TGFβ and PDL-1)	Weekly paclitaxel	Recurrent/metastatic GC	Not yet recruiting
**NCT03539822 Phase I/II**	Cabozantinib plus Durvalumab (GC cohort)	N/A	Advanced GC, GEJC, others	Recruiting
**NCT04164979 Phase II**	Cabozantinib Combined with Pembrolizumab	N/A	Metastatic or recurrent GC, GEJC (progressed, or not tolerated, at least one prior line of chemotherapy)	Recruiting
**NCT04082364** **Phase II/III**	Margetuximab Retifanlimab, Tebotelimab, Trastuzumab	XELOX or mFOLFOX6	HER2+ GC or GEJC	Active, not recruiting

**Table 2 cancers-14-05940-t002:** Ongoing Clinical Trials Featuring NK Cell Therapy in Gastric Cancer Patients.

Clinical Trial Information	NK Cell Type	Other Therapy	Patient Population	Status
**NCT03319459** **Phase I**	Fate-NK100 (allogeneic NK cell subset expressing the maturation marker CD57)	N/A	Various solid tumors	Completed, awaiting results
**NCT05069935** **Phase I**	FT538 NK cells derived from an induced pluripotent stem cells with modifications to enhance ADCC and persistence	All receive cyclophosphamide and fludarabine prior to NK cells. Plus: Avelumab in FDA-approved tumors, Trastuzumab in HER2+	Various solid tumors	Recruiting
**NCT04319757** **Phase I**	ACE1702: anti-HER2 antibody-cell conjugate, off-the-shelf NK cell product	Cyclophosphamide and fludarabine prior to NK cells.	Advanced or metastatic HER2+ tumors	Recruiting
**NCT04385641**	Umbilical cord blood derived NK cells	Cyclophosphamide and fludarabine prior to NK cells.	Advanced GC or GEJC	Recruiting
**NCT04847466** **Phase II**	Irradiated PD-L1 CAR-NK Cells	Pembrolizumab and N-803 (IL15 super-agonist)	Recurrent or metastatic GC or head/neck cancer	Recruiting
**NCT05207722** **Phase I/IIa**	CYNK-101: NK cells from human placental CD34+ cells, altered to express cleavage-resistant CD16	Induction with Pembrolizumab, Trastuzumab and a Fluoropyrimidine/Platinum based Chemotherapy regimen	Locally Advanced Unresectable or Metastatic HER2+ GC or GEJC	Recruiting
**NCT02839954** **Phase I/II**	anti-MUC1 CAR-NK cells	N/A	MUC1+ advanced refractory or relapsed solid tumors	Unknown

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
