# Peer review of "Gastric Cancer and the Immune System: The Key to Improving Outcomes?"

_cancers, 2022, doi:10.3390/cancers14235940_

Round 1
Reviewer 1 Report
Overall, the whole structure of this study is good and some corrections are recommended for providing clear information. Particularly, I listed the following comments in detail here.
Some sentences lack of references. For example, “Many GC patients present with advanced or metastatic disease due to the insidious nature of common presenting symptoms such as dyspepsia, anorexia, weight loss, and abdominal pain. This delay in presentation contributes to a high mortality rate.”, “These distinct categories display different anatomical predilections, along with trends towards development in older vs. younger patients”,
Additionally, all of the names and terms should be completely mentioned for the first time. For example, H. pylori.
Precise conclusion as it’s too short in its current form. Hence, add a significant statement that must be structured as “what was offered by authors? Do the authors have more thoughts on this field?
Author Response
Reviewer 1:
Comment 1: Some sentences lack of references. For example, “Many GC patients present with advanced or metastatic disease due to the insidious nature of common presenting symptoms such as dyspepsia, anorexia, weight loss, and abdominal pain. This delay in presentation contributes to a high mortality rate.”, “These distinct categories display different anatomical predilections, along with trends towards development in older vs. younger patients”
Response to comment 1: added citations to all sentences described above.
Comment 2: Additionally, all of the names and terms should be completely mentioned for the first time. For example, H. pylori.
Response to comment 2: Ensured that all terms are written out in full the first time they are used.
Comment 3: Precise conclusion as it’s too short in its current form. Hence, add a significant statement that must be structured as “what was offered by authors? Do the authors have more thoughts on this field?
Response to comment 3:
Conclusion: further input on the future of the field was added to this section.
Thank you for your comments and time spent reviewing.
Reviewer 2 Report
I read the manuscript which is well written and very detailed with up to date information. It could be published as written.
This is a review of immunotherapy for gastric cancer, the topic very relevant has immunotherapy advances help gastric patient surviving longer. It is a nice succinct review that encompasses all newly published study data. Regarding the methodology, authors could have added how they sorted the articles they reviewed and what search engine(s) they looked at (pubmed, clinicaltrials.gov, etc...).
The conclusions consistent with the evidence and arguments presented and they address the main question posed, references are appropriate and tables are very detailed and informative.
Author Response
Reviewer 2:
Comment 1: Regarding the methodology, authors could have added how they sorted the articles they reviewed and what search engine(s) they looked at (pubmed, clinicaltrials.gov, etc...).
Response to comment 1: Added a sentence on resources used (PubMed and ClinicalTrials.gov) in the intro.
Thank you for your comments and time spent reviewing.